# Insights into Muscle Contraction Derived from the Effects of Small-Molecular Actomyosin-Modulating Compounds

**DOI:** 10.3390/ijms232012084

**Published:** 2022-10-11

**Authors:** Alf Månsson, Dilson E. Rassier

**Affiliations:** 1Department of Chemistry and Biomedical Sciences, Linnaeus University, 391 82 Kalmar, Sweden; 2Department of Kinesiology and Physical Education, McGill University, Montreal, QC H2W 1S4, Canada

**Keywords:** myosin, actin, myosin-active compounds, muscle contraction, mechanokinetic model, statistical model

## Abstract

Bottom-up mechanokinetic models predict ensemble function of actin and myosin based on parameter values derived from studies using isolated proteins. To be generally useful, e.g., to analyze disease effects, such models must also be able to predict ensemble function when actomyosin interaction kinetics are modified differently from normal. Here, we test this capability for a model recently shown to predict several physiological phenomena along with the effects of the small molecular compound blebbistatin. We demonstrate that this model also qualitatively predicts effects of other well-characterized drugs as well as varied concentrations of MgATP. However, the effects of one compound, amrinone, are not well accounted for quantitatively. We therefore systematically varied key model parameters to address this issue, leading to the increased amplitude of the second sub-stroke of the power stroke from 1 nm to 2.2 nm, an unchanged first sub-stroke (5.3–5.5 nm), and an effective cross-bridge attachment rate that more than doubled. In addition to better accounting for the effects of amrinone, the modified model also accounts well for normal physiological ensemble function. Moreover, a Monte Carlo simulation-based version of the model was used to evaluate force–velocity data from small myosin ensembles. We discuss our findings in relation to key aspects of actin–myosin operation mechanisms causing a non-hyperbolic shape of the force–velocity relationship at high loads. We also discuss remaining limitations of the model, including uncertainty of whether the cross-bridge elasticity is linear or not, the capability to account for contractile properties of very small actomyosin ensembles (<20 myosin heads), and the mechanism for requirements of a higher cross-bridge attachment rate during shortening compared to during isometric contraction.

## 1. Introduction

Muscle contraction results from interactions between billions of myosin motors and actin molecules. These proteins are located in thick and thin filaments, respectively, in a highly ordered 3D lattice in the muscle sarcomere. The approximately 2 µm-long sarcomeres are connected in series in 1–3 µm-wide myofibrils that fill the muscle cells. As a result of the interactions between myosin and actin, the thin and thick filaments slide past each other at velocities of up to tens of micrometers per s as a result of nm displacements produced by actin–myosin cross-bridges. The summation of the shortening of all sarcomeres in series along the myofibrils causes the muscle cell to shorten by appreciable distances. Additionally, by summation of the forces in all half-sarcomeres over the muscle cross-section, the pN forces produced by each myosin cross-bridge add up to muscle-produced forces corresponding to up to 1000 kg or more.

With the described hierarchical organization of the muscle, the contractile properties directly reflect the mechanical and kinetic properties of the cross-bridges. This allows the use of computational models to simulate muscle contraction, as pioneered by Huxley [1] and later formalized by Hill and Eisenberg [2,3,4]. In these models, properties such as the muscle force–velocity relationship, (ATP)–velocity relationship, and ATP turnover rate vs. velocity, among other average properties, reflect the probability of different actomyosin cross-bridge states. These properties can be calculated using systems of differential equations in the state probabilities. As has been demonstrated recently [5,6,7,8,9], this allows for very good predictions of the physiological properties of muscle under full (or close to full) activation. If the model parameter values (cross-bridge stiffness, strain-dependent transition rate constants, etc.) are from studies of isolated actin and myosin, e.g., single-molecule studies and solution biochemistry, we use the term bottom-up models (see also [9,10]). The success of such models in predicting muscle function essentially from single-molecule properties suggests that the predicted phenomena are affected by neither cooperative interactions in the large ordered actomyosin ensembles, nor by the presence of accessory, regulatory proteins. Bottom-up models are also of potential value in drug discovery. They thus allow predictions of ensemble behavior and muscle function in the presence of drug candidates based on initial studies of isolated proteins. This would be useful for evaluating if desired drug effects are expected in a real muscle without actually expanding the experiments to such preparations, with benefits from both an ethical and a cost perspective. However, in order to use the models for such purposes, it is important that they not only accurately predict physiological muscle properties but also drug effects, mutation effects, etc. In order to allow effective testing, it is required that the drug and/or mutation effects be well-characterized on the single-molecule level with minimal ambiguity, allowing the input of well-defined parameter values. Moreover, detailed quantitative characterization of the muscle properties upon treatment with a drug or the presence of a given mutation need to be available for meaningful evaluation of the predictive power. 

One small molecular compound that fulfills the above requirements is amrinone. This compound was introduced as a phosphodiesterase inhibitor with the aim of treating heart failure [11]. Later, it was found that it has well-defined effects on both frog and mammalian muscles [12,13,14] via direct actions on myosin [15,16]. Its molecular effects, attributed to the inhibition of strain-dependent ADP release [16], led to several changes in the force–velocity (FV) relationship with increased maximum isometric force (*F*_0_), reduced the maximum velocity of shortening (*V*_0_), reduced the overall curvature of the relationship, and reduced the deviation of the FV relationship from a hyperbola at high loads [12,13,15,16]. Whereas some of these effects on the FV relationship have been predicted by recent models, the quantitative goodness of fit has been variable [16,17]. It is therefore of value to analyze the effects of amrinone in greater detail. Other small molecular compounds whose mechanisms of action have been quite well-characterized are omecamtiv mecarbil (OM) [18,19,20,21,22,23,24,25], initially introduced as a myosin activator in heart failure [26,27], and the blebbistatin family of myosin-inhibiting compounds [6,28,29,30,31,32,33,34,35]. The latter are used as myosin inhibitors in cell studies [28] but have also been evaluated as muscle relaxants [36]. With respect to the molecular mechanism, OM increases the rate of Pi release while inhibiting the power stroke [18,19,20,21,22,23,24,25]. Blebbistatin, on the other hand, inhibits Pi release [29,31] an effect that was recently [6] interpreted as being due to the inhibition of the transition from a pre-power stroke state, AMDP_PP_, into a Pi-release state, AMDP_PiR_ (cf. Figure 1A). This would shift the rate-limiting step for the actin-activated ATPase of myosin from cross-bridge attachment to the AMDP_PP_ state to the AMDP_PP_–AMDP_PiR_ transition. Whereas the effects of OM and blebbistatin are well-characterized on the molecular level, the details of their effects on the FV relationship are not. However, blebbistatin appreciably reduces both *V*_0_
*and F*_0_ [6,30,37] and is dependent on regulatory light chain phosphorylation to have an effect on velocity in muscle cells [37]. OM also appreciably reduces *V*_0_ while only slightly reducing *F*_0_ during full calcium activation [38].

Here, we test the recently developed bottom-up model [6] (defined in Figure 1 and Table 1 and Table 2) with respect to its capacity to account for changes in the FV relationship of muscle produced by the mentioned myosin-active small molecular compounds based on molecular mechanisms. Our results demonstrate that overall, the model produces good predictions of the FV data based on the major molecular mechanism for each compound. However, the model [6] suffers similar weaknesses as one other recent models [17] with respect to its quantitative reproduction of the effects of the drug amrinone on F_0_ and V_0_. Interestingly, the further constraints on the models by the drug effects together with the availability of new experimental data from isolated molecules allow us to improve the model to overcome the mentioned shortcomings while still accounting for other data. Finally, we develop a Monte Carlo simulation-based version of the model that allows its use with small myosin ensembles. Specifically, we test the latter version of the model with regard to its capability to account for experimental FV relationships at varied MgATP levels in small isolated actomyosin ensembles. We discuss remaining outstanding issues related to challenges to obtain reliable parameter values due to experimental uncertainties and differences between labs. Additionally, our study is relevant for understanding the operation of actin–myosin molecular motor ensembles, as it focuses on key issues with currently diverging views. In particular, this includes the possibility of faster cross-bridge attachment rates during active shortening compared to during isometric contraction [39,40,41] and the possibility of non-linear cross-bridge elasticity in muscle cells [8,42,43,44,45]. 

## 2. Results

### 2.1. Simulation of Force–Velocity relationships under Physiological Conditions 

We simulated the FV relationship for muscle in the absence of small molecular compounds using a model from [6] (Figure 1) that was slightly modified with respect to parameter values (Table 1 and Table 2). The results are compared to experimental data from living mouse toe muscle at 30 °C (reproduced from [47] in Figure 2), similar to the temperature at which the model parameters (Table 1 and Table 2) were derived. It can be seen (Figure 2A) that the simulated V_0_ value is low compared to the experimental values and are in the range of 13,000–18,000 nm/s ([13,48,49]; reviewed in [8]). We attribute this to either of the following factors or a combination of them: First, the experimental results in Figure 2 are from mouse toe muscle, which is expected to have a velocity at the fast end of the range, as it is a fast muscle from a small animal [50]. These data [47] were used because they are particularly complete in the high force range. However, most of the model parameter values (Table 1 and Table 2), particularly those of relevance for *V*_0_, are from fast rabbit psoas muscles [43,51]. A second possible reason for a low *V*_0_ in the modelling is that a linear myosin cross-bridge elasticity is assumed, whereas it is possible that the real cross-bridge elasticity is non-linear [8,42,43,52]. This issue is discussed further below. With regard to *F*_0_ in model and experiments, the numerical values cannot be directly compared due to experimental complexities related to the use of a whole-muscle preparation in the experiments, e.g., presence of appreciable extracellular space, non-parallel muscle fibers, failure to activate some fibers and, finally, intracellular space between myofibrils. Despite the complexities, it can be seen in Figure 2B that the shape of the FV relationship is well-predicted by the model. This particularly applies to the general curvature, but it also includes the deviation of the relationship from a single hyperbola at high loads. 

### 2.2. Effects of Small Molecular Compounds on the Force–Velocity Relationship

A model of general validity should account for the key contractile effects of mutations, drugs, and altered experimental conditions in addition to physiological data. This capability was tested for our standard model by investigating its predictions for the effects of the drugs amrinone, blebbistatin, and OM on the FV relationship. For amrinone, the central mechanism seems to be inhibition of the strain-dependent transition prior to ADP release [16]. The implementation of this idea by reducing Δ*G_AMDH-AMD_* from 2 to 0.5 k_B_T in the model (cf. Figure 2B) predicts a substantial increase in *F*_0_ simultaneously with a reduction in *V*_0_ (Figure 2A). These results, reflecting near-saturating amrinone concentrations of 1–2 mM, are qualitatively similar to the experimental findings. However, the predictions of the model in Figure 2A as well [6] as those of some other models (based on similar assumptions) [17] differ from the experimental results with amrinone in several respects: 1. appreciably higher increase in *F*_0_ in the model; 2. lower reduction in *V*_0_; 3. increased, rather than decreased, curvature of the FV relationship; and 4. limited effects of amrinone on the deviation of the FV data from a hyperbola in the model. Such differences between model and experimental results were smaller in earlier model simulations (particularly [16]). These issues are considered below, where we also describe the results of an optimized version of the model. 

With regard to blebbistatin, we recently [6] attributed its contractile effects and the inhibition of Pi release to a greatly reduced transition rate between the pre-power-stroke state (AMDP_pp_) and the Pi-release state (AMDP_PiR_). We then implemented this idea in a mechanokinetic model [6]. The standard model that we use here increased the value of k_Pr+_’ from 1000 s^−1^ to 3000 s^−1^ but is otherwise identical to the model in [6]. It is therefore no surprise that a decrease in k_Pr+_’ from 3000 s^−1^ to 5 s^−1^ (cf. [6]) accounts for the quantitatively larger reduction in V_0_ than in F_0_, as seen experimentally with blebbistatin [13]. 

For OM, molecular mechanistic studies [21,22,53] suggest that it stabilizes an actomyosin pre-power-stroke state, which is associated with an increased rate of the transition into this state (the Pi-release state in [46]) and with appreciable slowing of the subsequent power stroke. We bluntly implemented these ideas in our standard model by increasing the free energy difference between the AMDP_pp_ and AMD_L_ states from 1 to 6 k_B_T while reducing the free energy difference between the AMD_L_ and the AMD_H_ states (the power-stroke transition) from 14 to 0 k_B_T. These changes in the model led to a substantial reduction in V_0_ and only minor changes in F_0_, similar to observations in experiments on human ventricular muscle tissue and isolated myosin [23]. 

Finally, for lowered [MgATP], similar to amrinone, the model predicted increased *F*_0_ and reduced *V*_0_, which is broadly in agreement with experimental findings [54,55]. For varied [MgATP], blebbistatin, and OM, no complete FV data are, to the best of our knowledge, available from studies of intact muscle fibers. FV data are available for varied [MgATP] levels using skinned fibers from rabbit psoas [55] and frog muscle [54]. Similar data from skinned fibers are available for 20 µM blebbistatin [37], where, however, effects on velocity are only observed under conditions with phosphorylated myosin-regulatory light chains. The skinned fiber FV data usually exhibit greater variability in the exact shape of the relationship, and they reveal less details than the intact fiber data. For varied [MgATP], blebbistatin, and OM, we therefore only discuss the effects of the intervention on *V*_0_ and *F*_0_ in the analyses below and not the shape of the FV relationship (despite showing simulated FV data). 

In summary, key effects on the FV relationship of the small molecular compounds amrinone, OM, and blebbistatin as well as lowered [MgATP] are reasonably well accounted for by our standard model if parameter values are changed on basis of the molecular effects of the compounds suggested by analyses on isolated proteins. However, we also note quantitatively poor predictions of the detailed effects of amrinone on the FV relationship. We consider this phenomenon further because detailed experimental FV data from intact muscle fibers exist for the effects of 1–2 mM amrinone. 

### 2.3. Towards an Optimized Model

A clue to a deficiency in our present model is obtained by inspecting differences from a model from Albet-Torres et al. [16] that was better at quantitatively reproducing the effects of amrinone on the FV relationship. One key difference of that model from the present version was a larger second sub-stroke of the power stroke (2.2 nm instead of 1 nm) associated with the transition from AMD_H_ to AM/AMD. 

Whereas there were also other differences, the larger amplitude of the second sub-stroke is of particular interest because such a larger second sub-stroke (~2.5 nm) was also suggested by recent single-molecule data [44] using full-length myosin incorporated into filaments. This differed from estimates obtained using isolated myosin subfragment 1 (~1 nm) [56], which we employed in our previous modelling work but also in the present Figure 2. In order to evaluate the importance of the amplitude of the second sub-stroke, we modeled the FV relationship and the effect of amrinone for different amplitudes of this second sub-stroke in the range of 1–3.5 nm while keeping all other parameter values constant. We also repeated this analysis under three different conditions, all with Δ*G_AMDL-AMDH_ +*Δ*G_AMDH-AM/AMD_* = 16 k_B_T but with Δ*G_AMDL-AMDH_/*Δ*G_AMDH-AM/AMD_*, assumed to be either 14 k_B_T/2 k_B_T, 12 k_B_T/4 k_B_T, or 10 k_B_T/6 k_B_T under physiological conditions. The amrinone effects were then simulated by changing these ratios to 14 k_B_T/0.5 k_B_T, 12 k_B_T/1 k_B_T, or 10 k_B_T/1.5 k_B_T, i.e., by reducing Δ*G_AMDH-AM/AMD_* to 25 % of the physiological value. The effects of these changes in the model parameters on the FV relationship are summarized in Figure 3A–D, where the second sub-stroke amplitude, *d*_2_, is related to the varied parameter value *x*_2_, such as *d*_2_
*= x*_2_*-x*_3_ with x_3_ constant at 7.7 nm. Additionally, full simulated FV data sets are shown in Figure 3F after changes in the model parameters to *x*_2_ = −5.2 nm with Δ*G_AMDH-AM/AMD_* = 12 k_B_T and Δ*G_AMDL-AMDH_* = 4 k_B_T under physiological conditions and an assumed change of Δ*G_AMDL-AMDH_* to 1 k_B_T in the presence of 1–2 mM amrinone. It is clear from this analysis that the fractional increase in the second, compared to the first, sub-stroke in the model leads to better quantitative reproduction of the effects of amrinone on *F*_0_ and *V*_0_. However, the changes in the model parameters (i.e., change of *x*_2_ from −6.7 to −5.2 nm) also result in curvature of the FV relationship that is too high and with lower maximum power output than observed experimentally. This effect leads us back to a path that has been tread before, i.e., the idea that the apparent rate of cross-bridge attachment may be higher during muscle contraction associated with length changes than during isometric contraction [39,40,57].

Most simply, the steady-state FV relationship with a higher attachment rate constant during shortening can be simulated by assuming a higher cross-bridge attachment rate at all loads. An increase in *k_on_’* from 130 to 325 s^−1^ could account for the high power output during shortening against intermediate loads rather well, with only minimal effects on the steady-state isometric force (Figure 4A). However, the experimentally observed deviation from a hyperbola of the FV relationship at high loads is not very well-predicted with this set of parameter values. The non-hyperbolic deviation has previously been shown to reflect the positions along the x-axis (cf. Figure 1B) of the free energy minima of the different cross-bridge states [40,60] as well as the level of these free energy minima relative to each other [40]. Here, we modified the positions slightly from those used in Figure 3F and Figure 4A in order to improve the predictions. This can be justified within the bottom-up modeling framework due to uncertainties of up to 1 nm [6]. First, we tested to increase the values of *x*_1_ and *x*_11_ (Figure 4A–D), thereby increasing the total stroke following attachment from about 7 to 8 nm, well within current estimates (reviewed in [10]). As predicted based on previous results, the shape of the high-force region of the FV relationship is sensitive to small modifications of *x*_1_ and *x*_11_. However, the small modifications did not fully accommodate the experimentally observed deviation of the FV relationship. To achieve this, we also fine-tuned the value of x_2_ from −5.2 nm to −5.5 nm (Figure 4E), showing that the exact non-hyperbolic shape is also quite sensitive to this parameter. As demonstrated previously [9], we found that an increase in *x*_1_ also reduces *F*_0_ (data not shown) while increasing the maximum power output. 

Based on the above analysis and the renewed evidence suggesting the need to assume a velocity-dependent attachment rate constant, we updated our standard set of parameter values (for physiological conditions) to those given in Table 3. It is of interest to note that the changes in *x*_1_, *x*_11_*,* and *x*_2_ are in good agreement with new single-molecule data for the power stroke distances [44,45].

### 2.4. Prediction of Ensemble Contractile Function with and without Small Molecular Compounds Using Optimized Model

Using the updated parameter value set (Table 3, Figure 5, and its legend), we re-ran simulations of the FV data in the presence of amrinone, varied [MgATP], blebbistatin, and omecamtiv mecarbil (Figure 6). Here, we assumed that that parameter values were changed by the small molecular compounds in a similar way as in the simulations in Figure 2, as specified in the legend of Figure 6. We found that with the optimized model, not only the physiological steady-state data, are well accounted for. Additionally, the amrinone effects are in closer agreement with the experimental findings. Thus, V_0_ was predicted to be reduced by 25 % in the model compared to between 25 and 35% in different experimental preparations from frog, mouse, and rabbit myosin/muscle fibers with 1–3 mM amrinone [12,13,15,16,17]. The maximum force, *F*_0_, on the other hand, was predicted to be increased by 16% in the model compared to by 4%, 14%, and 8% in experiments using fast mouse[13], rat [61], and frog[12] skeletal muscle, respectively. If *x*_2_ = −5.2 nm (instead of −5.5 nm, as in the final optimized model), the predicted reduction in velocity was 27%, and the increase in force was 11%, in even better quantitative agreement with the experimental data. However, independent of which of these values of *x*_2_ was used, there was a minimal effect of amrinone on the overall curvature of the FV relationship in the model. This is different from experimental data that always showed reduced curvature in the presence of amrinone. On the other hand, the change of going from the previous standard to the new optimized model is in the right direction because the use of the previous standard parameter values (Table 1 and Table 2) predicts an increased curvature upon the addition of amrinone. Furthermore, the optimized model suggests reduced deviation in the FV relationship from a hyperbola (Figure 6); however, this is not reflected in the *F*_0_*/F*_0_*** ratio, as is usually the case. 

We also implemented a Monte Carlo simulation approach to enable studies involving small myosin ensembles, e.g., expressed proteins isolated from cell systems. The validity of this implementation is supported by the results in Figure 7, showing that simulated force–velocity data of large myosin–actin ensembles are virtually identical using the Monte Carlo approach (see output of simulations in Appendix A) and the solution of differential equations in state probabilities. Using the Monte Carlo version of the model, we also showed that the key predicted consequences of reducing the myosin ensemble size are a reduced maximal (unloaded) shortening velocity with negligible effects on the maximum isometric force normalized to the number (*N*) of available myosin heads (Appendix A). This expands previous data for small and large ensembles using less optimized models with fewer states [62,63].

In accordance with experimental data [44,45,64,65,66,67,68], the simulated FV relationship for moderately small ensembles (*N*, 33–39; Figure 7) exhibits the typical Hill shape [69]: an overall hyperbolic shape. However, the model predicts a less curved relationship for the low-ensemble case, consistent with one [44] but not other sets [66,70] of small-ensemble near-physiological (mM) [MgATP] experimental data. Notably, we could not derive the typical Hill relationship (see Materials and Methods) in our simulations when *N* was reduced to 16–19 heads (Figure 7), corresponding to a maximum isometric force of about 20 pN. This is in contrast to the situation in the experiments of Pertici et al. [66], where a low number of myosin heads (*N* = 16) with an isometric force of 20 pN (similar to our simulations) produced a typical Hill-style FV relationship. Previously [64], a Hill-style FV relationship was also found for even fewer myosin heads (8) but at <100 µM MgATP. It can further be seen in the simulations in Figure 7 (particularly Figure 7B) that the non-hyperbolic shape of the FV relationship for *N* = 33–39 at a high load is largely concealed by stochastic noise. More generally, the Monte Carlo simulations suggest that noise effects would make it quite challenging to derive accurate FV relationships from small ensemble measurements. In the simulations, this is attributed both to variability within a given simulation run and variability between runs (Figure 8; Appendix A). With regard to the between-run variability, this is particularly severe for simulations of isometric contraction. The effect is associated with the single-site assumption, where only one short segment (approximately 5 nm) of 36 nm along the actin filament is available for myosin head attachment, corresponding to a probability of *p* ≈ 5/36 ≈ 0.14. We show in the Supplementary Information that under our simulation conditions, this is expected to lead to number of attached heads *N_at_*_t_ that is binomially distributed *N_att_* ∈ Bin (*N*, *p*) between runs, with a mean value *Np* and standard deviation $\sqrt{Np\left(1-p\right)}$ , i.e., with a signal-to-noise ratio associated with the variability between runs $proportional\;to\;\sqrt{\left\{Np/\left(1-p\right)\right\}}$. Clearly, both an increase in *N* and an increase in p (as with several actin sites per target zone) would increase this signal-to-noise ratio, consistent with the changes in the signal-to-noise ratio between simulations for *N* = 16–19 and *N* = 33–39 (Figure 8). The between-run variability was markedly reduced during the simulated slow shortening of amplitude >36 nm, which we attribute to averaging out of the effect of the non-uniform distribution of myosin heads. In the present analysis, we largely eliminated the effect of the in between-run variability in isometric force by averaging over a large number of runs. The remaining within run variability was higher during simulated shortening than isometric contraction. This effect was not only attributed to stochastic attachment and detachment events, but also, the cyclic increase in cross-bridge attachment as the number of cross-bridges available for attachment varies. The effect of this variability was largely eliminated by averaging all force data during a simulated iso–velocity-shortening run producing low remaining variability (Figure 8).

The simulations in Appendix A depict force responses to iso–velocity shortening of an actin filament propelled by *N* = 16–19 or *N* = 33–39 available myosin heads. Ramp shortening rates of approximately 5, 30%, or 60% of *V*_0_ were imposed after the attainment of steady-state isometric force. The average isometric force obtained for these conditions (averaged both between and within runs) was 22.9 ± 5.2 pN (mean ± 95% CI; *N* = 16–19 myosin heads; 28 runs; 16–66 data points per run) and 51.6 ± 6.6 pN pN (*N* = 33–39 myosin heads; 30 runs, 15–55 data points per run), respectively (Figure 8). This may be compared to experimental data with an isometric force of about 20 pN for a myosin ensemble of 16 heads [66] and 40 pN for 17 heads [44]. The comparison of these results with the simulated data in Figure 7 and Figure 8 suggest that double the number of heads are attached in the experiments compared to in the simulations for a given *N*, consistent with approximately up to two accessible sites per actin target zone in experiments compared to one site per target zone in the simulations. In contrast to the very low signal-to-noise ratio in the simulations for 17 heads, the experimental data [66] with similar isometric force (for 16 heads) exhibited a high signal-to-noise ratio.

As mentioned above and discussed further below, our Monte Carlo simulations experienced challenges in accounting for the behavior of small myosin ensembles of about *N* = 20 and below (corresponding to forces of around 20 pN and below). However, the behavior of larger ensembles seems well predicted. This allowed us to subject the Monte Carlo version of the optimized model to a further test with regard to its prediction of effects of varied [MgATP] on FV data obtained in experiments using isolated actin and myosin filaments [70]. Based on the isometric force in these experiments of approximately 120 pN, we estimated the number of available heads to be *N* ≈ 100. The temperature in the experiments was approximately 22 °C, suggesting changes in the model parameters, as in Table 3. Simulations of FV data at 1.2, 0.5, and 0.1 mM MgATP on these assumptions are shown together with experimental data for 1.2 and 0.5 mM MgATP in Figure 9. These data show that the predicted FV relationship, with parameter values adjusted as in Table 3 and [MgATP] and *N* set to 1.2 mM and 100–110, corresponds reasonably well to the experimental data. The small discrepancy in the predicted *V*_0_ from the experimental value can be attributed to the choice (based on pilot simulations) of somewhat low *N* values in the simulations. Increased *N* would increase both the total isometric force (not *F*_0_/*N* though) and V_0_ (Appendix A). A higher *V*_0_ value would also be expected if the cross-bridge elasticity had been assumed to be non-linear. Lowering [MgATP] to 0.5 mM results in the prediction of reduced maximum velocity, reduced curvature of the FV relationship, and increased F_0_, just as shown in the large ensemble simulations in Figure 6. The predictions for the changes in *V*_0_ and the FV curvature with reduced [MgATP] from 1.2 to 0.5 mM in the simulations are qualitatively similar to the experimental findings but are appreciably smaller in magnitude. A better reproduction of the experimental data at 0.5 mM MgATP is achieved by lowering [MgATP] to 0.1 mM in the simulations.

The lower curvature of the FV relationship at 1.2 mM MgATP that we observed in the simulations compared to the experimental data in Figure 9 may be related to the tendency of our small ensemble simulations to predict lower curvature of the relationship (Figure 7). However, it is also important to keep in mind the substantial variabilities between labs in terms of force–velocity data for both isolated proteins and muscle cells (Figure 10).

A discrepancy between the experimental data and model predictions that is worth noting is an appreciably lower *F*_0_ at lowered [MgATP] levels in the experiments (see legend of Figure 9 and [70]). In contrast, the model predicts increased *F*_0_ with reduced [MgATP], both in the large ensemble version in Figure 6 and in the small ensemble (Monte Carlo) version in Figure 9. These predictions are consistent with previous results where the effects of varied [MgATP] were studied using skinned muscle cells [54,55]. For instance, in the study of Cooke and Bialek [55] using glycerinated rabbit psoas fibers, the force increased after lowering MgATP from 1 mM to 50 µM, where a maximum was observed for F_0._ Further reductions in [MgATP] led to a drop in F_0_. Presumably, the basis for the opposite effects in the filament experiments in Figure 9 is the way the isometric force is attained via the appreciable sliding of the filaments past each other. It is possible that this sliding may be partly inhibited by the lowered [MgATP] level to prevent attainment of the true *F*_0_.

## 3. Discussion

### 3.1. Summary and Main Implications

We refined a bottom-up model [6] that was previously found to account for a range of physiological phenomena in muscle contraction as well as the effects of blebbistatin by applying constraints based on contractile effects of other small molecular compounds. This model, whether implemented by solving differential equations in state probabilities or by Monte Carlo simulations (Materials and Methods), brings new insights and introduces the potential use of bottom-up models in predicting ensemble effects of drugs and myosin modifications in diseases. We show that the model is applicable to the simulation of experiments both on muscle cells and on small actomyosin ensembles in vitro. An important finding from our analysis is that changes in the parameter values that determine the amplitudes of the two sub-strokes of the power stroke have key roles in determining the shape of the FV relationship. Making these values more similar to those obtained in recent single-molecule studies (5.5 and 2.5 nm, respectively) [44] allows for better quantitative reproduction of the amrinone effects on *F*_0_ and *V*_0_. However, associated with these changes in the model, we found it necessary to increase the rate constant of cross-bridge attachment more than two-fold in order to account for the maximum power output. This means that a higher cross-bridge attachment rate is required to account for the maximum power (>300 s^−1^) than suggested by the rate of rise of isometric contraction (~130 s^−1^) and the maximum actin-activated ATP turnover rate (cf. [6]). This is in contrast to our recent modelling efforts assuming a short second sub-stroke of 1 nm (with a first sub-stroke of 5–6 nm; [5,6,7,9,71]). However, the finding is similar to previous data from models and experiments by us and others [1,39,40,57,72,73], as further discussed below.

### 3.2. Comparison with Previous Studies

As mentioned above, the amrinone effects were quite well-accounted for quantitatively by an earlier model [16]. However, it is difficult to directly compare the present results to those findings. First, the previous model was not stringently defined with respect to the relationship between the mechanical and biochemical (ATP turnover) states. Second, it [16] was applied particularly to data from frog muscle fibers but with parameter values obtained in rather incoherent ways by mixing data from frog muscle with other data extrapolated from experiments on isolated actin and myosin from mammalian muscle. Since then, our models have evolved in different ways, starting in [17] and [5]. First, we arrived at a one-to-one correspondence between mechanical and chemical states, and second, we switched from modelling fast frog muscle data to modelling fast mammalian muscle results using parameter values from fast mammalian actomyosin only (more details in [9]).

The bottom-up modeling approach deserves some general considerations. The key strategy is to stick with model parameters obtained in independent experiments on isolated proteins when simulating ensemble contractile data of muscle and to not change the parameter values to improve the fits, unless absolutely necessary. This approach means that we interpret any differences that emerge between experiments and the model as either being due to experimental uncertainties or as being due to the effects of cooperative effects in the actomyosin ensemble or of accessory proteins, etc., in muscle. Here, we take cooperative effects to mean that different kinetic properties that need to be assumed for each actomyosin interaction in an ordered myosin ensemble than for an isolated single molecule. In our most recent modeling efforts conducted before this study, we found that experimental uncertainties seem sufficient to explain deviations between experimental data and data predicted from bottom-up models [9,10,71]. However, the present model refinements, constrained by drug effects and consistent with recent single molecule experiments [44], has forced us to change our conclusions. This issue is of fundamental importance for understanding the operation of the actomyosin molecular motor system and is discussed in further detail in the next section.

### 3.3. The Need to Assume Higher Cross-Bridge Attachment Rate during Shortening

The high power output of a real muscle could not be accounted for by our optimized model without appreciably increasing the attachment rate constant to a value higher than that accounting for the maximum rate of rise in the isometric force and actin-activated ATP turnover rate. In order to account for this finding, it seems necessary to assume cooperative effects with a higher cross-bridge attachment rate during shortening than predicted from actomyosin kinetics derived from isolated proteins (and from the rate of rise of isometric contraction). In our recent models [5,6,7,8,9], where we assumed a shorter second sub-stroke, there seemed to be no need to invoke cooperative effects in order to explain the experimental data. This change in model properties, due to rather small alterations in *x*_1_*, x*_11_*,* and *x*_2_, demonstrates an appreciable sensitivity of the power output to these parameters. Furthermore, our analysis shows that small changes in these parameters also strongly affect the exact shape of the non-hyperbolic deviation of the FV relationship from a hyperbola at high loads, emphasizing and expanding previous findings [40,60].

A higher apparent cross-bridge attachment rate during shortening than during isometric contraction is intriguing. It is highly unlikely that the rate of cross-bridge attachment should change as the thin and thick filaments start to slide past each other during shortening. One may, however, consider the effect in relation to the so-called regeneration of the power stroke following a length step. Additionally, this process exhibits a faster rate constant than for cross-bridge attachment during isometric contraction and actin-activated ATP turnover [74]. Regeneration is defined as the process whereby the tension responses to a length step recover following a previous first length step to become similar to the tension responses (time course and amplitude) seen after that first step [74]. Both the apparent increase in the attachment rate constant during shortening and the regeneration of the power stroke may reflect either of the following processes: (a) the sequential action of the two heads of myosin during shortening [39,40,75]; (b) the slippage of myosin cross-bridges between neighboring sites (separated by 5.5 nm) along the actin filaments [6,72,73]*;* (c) the rate limitation for attachment (partly) associated with transitions between detached states, allowing completion of this transition as myosin heads slide between neighboring sites during shortening and thus bypassing the rate limitation in that case [76]; or (d) a thick filament mechano-sensing mechanism [77] that leads to increased recruitment of cross-bridges during shortening against high and intermediate loads [78].

Which of these mechanisms (a-d), or which combination of them (if any), might operate is presently unclear. However, each of the mentioned mechanisms would contribute to explaining an apparent velocity dependence of the attachment rate constant with a faster attachment rate during shortening than in isometric contraction. Here, it is important to note that the entire steady-state FV relationship is well-predicted by also assuming the higher attachment rate constant during isometric contraction because this would only have minimal effects on the steady-state isometric force. Importantly, however, the high rate would not be compatible with transient phenomena such as the rate of rise of isometric force during the onset of a contraction or after a release followed by reinstated isometric conditions.

### 3.4. Monte Carlo Simulations

A Monte Carlo simulation approach is required for predicting and analyzing the contractile behavior of small ensembles of myosin motors, e.g., as studied in recent experiments using optical tweezers [44,45,64,65,66,67] and nanofabricated cantilevers [68]. Such studies allow the evaluation of FV and other ensemble contractile data for myosin motors expressed in (and purified from) cell systems (cf. [79,80]). These experiments are not possible using methods developed for myofibrils and muscle cells. Similarly, simulation of the results is not feasible using the large ensemble version of the model based on solving differential equations. For instance, with a low number of available cross-bridges, the cross-bridge attachment rate is expected to appreciably influence the velocity (cf. [81,82]), rather than the detachment rate constant, at high motor densities [6,62,83]. Monte Carlo simulations [5,6] allow us to take this and some other effects into account. The small ensemble experiments and the associated use of Monte Carlo simulations to analyze the data would be of value for investigating the effects of disease-causing mutations (e.g., in cardiomyopathies), ensemble drug effects, as well as mechanisms for contractile properties of myosin ensembles based on molecular characteristics.

After demonstrating the general validity of the Monte Carlo approach by comparing simulations of large ensemble data to those obtained by solving differential equations in state probabilities (Figure 7), we went on to demonstrate the main predictions of the model due to a reduction in the myosin ensemble size, *N*. First, we found that the model predicts that the force/available head is virtually unchanged, whereas the maximum velocity of shortening is appreciably reduced with reduced *N* (Appendix A). Moreover, we found that the simulation of data at reduced N is appreciably affected by variability between simulation runs and within a given run (Figure 8, Appendix A). We considered the basis for these types of variability above and found that the influence of noise seems to be appreciably worse in simulations than in experiments, particularly at the lowest *N* values studied here (*N* < 20). We also found that this effect seems to be associated with a failure of the model to predict a hyperbolic shape of the FV relationship in contrast to what was observed experimentally by Pertici et al. [66]. However, a tendency for a non-hyperbolic FV relation was seen in previous experiments [84] in which loads were imposed using electrostatic fields rather than via force feedback using optical tweezers.

Experimental FV data at physiological ATP are well-predicted (Figure 7) within the experimental variability (Figure 10) for a myosin ensemble with *N* > 33 and isometric force of approximately 50 pN. This seems generally consistent with the data of Cheng et al. [70] and Kaya et al. [44] (who assume two sites per actin target zone in their experiments). However, the failure of our model for *N* = 17–18 to predict a hyperbolic force–velocity relationship seems at odds with the experiments of Pertici et al. [66], clearly demonstrating a hyperbolic FV relationship that was well-fitted by the Hill equation (Equation (14)). Similar to the high signal-to-noise ratio in their experiment compared to the low signal-to-noise ratio in our simulations, the reasons are not entirely clear. It is possible that the low signal-to-noise ratios in our modelling is itself important in this context (see above) in relation to the single-site assumption and lack of damping as mentioned above. Moreover, we also assumed [MgATP] = 5 mM, ionic strength >100 mM, and temperature = 30 °C in our simulations, whereas Pertici et al. [66] performed their experiments at [MgATP] = 2 mM, an ionic strength of 80 mM, and a temperature = 23 °C. It is unclear whether these factors are sufficient to explain the difference between the experiments of Pertici et al. [66] and our simulations.

The otherwise satisfactory performance of the model for *N* > 30 allowed us to go on to evaluate effects of varied [MgATP] using isolated myosin and actin filaments with *N* appreciably greater than 30. In these studies, we found qualitatively similar effects of varied [MgATP] in the simulations compared to the experiments. However, the absolute values of the curvature of the FV relationship at the highest [MgATP] level was lower than in the experiments discussed above. Moreover, in order to predict the typical effects of lowering [MgATP] from 1.2 to 0.5 mM, we had to assume a reduction from 1.2 to 0.1 mM in the simulations. This finding is actually not unexpected from several previous studies of the relationship between [MgATP] and *V*_0_ predicting that the MgATP concentration (*K_M_*) for half-maximum unloaded velocity (i.e., *V*_0_) is several-fold lower than in experiments [5,8,17,85]. These previous studies also suggested that a way to amend this problem is to leave the assumption of a linear cross-bridge elasticity and to instead assume a non-linear cross-bridge elasticity of the type observed in [43]. This idea was not further tested here, but it is an important observation. We also noted that our model could not account well for the effect of lowered [MgATP] on maximum isometric force, but this seems to be due to the particular way that maximum isometric force was attained in the experiments, as considered in the Results section.

### 3.5. Limitations

Models are only approximations of the real world, and it is also important to realize that models are not better than the underlying experimental data that are compared to the model results and/or the data used to derive the values of the model parameter. It is well-known that such experimental data vary between labs and even between experimental conditions in a given lab. This issue is clearly illustrated by the comparison of different sets of FV data from both muscle and small ensembles of isolated proteins in Figure 10.

Moreover, there are also some conceptual limitations of the present model. One of these is the simplified single binding site assumption (often made in models), where only one myosin binding site is assumed to be available per 36 nm half-repeats of the actin filament and only one myosin head (out of the two per myosin molecule) is available for binding to this site. This assumption means that the number of available cross-bridges may be 2–3 times higher for geometrical reasons in real muscle than assumed here [7]. However, with the exception of reducing extensive properties such as maximum isometric force (e.g., leading to corrections in Figure 3A) and an ATP turnover rate that increased by 2–3-fold, this assumption, normally does not noticeably alter the prediction of intensive properties such as *V*_0_*,* the increase rate of force, and the shape of the force–velocity relationship [7]. In the present case, however, the assumption precludes realistic representation of the phenomena (listed as a-d above) that may underlie the apparently faster cross-bridge attachment rate during shortening than during isometric contraction. The only way to represent such effects in the present model is by a faster cross-bridge attachment rate. Furthermore, the assumption of a single site contributes to the variability when *N* is low.

A limitation of any model, presently, is that it is unclear whether the cross-bridge elasticity is linear or non-linear [7,8,42,43,44,86]. This is a critically important issue. If the cross-bridge elasticity is non-linear in muscle fibers in the way suggested by single molecule experiments [43] with very low stiffness at negative *x* values, *V*_0_ would be appreciably higher for a given set of rate constants. Indeed, it was already pointed out in [87] that the only ways to account for the high velocity of shortening is to either to appreciably assume increased detachment rate constants for cross-bridges at negative strain or to assume non-linear cross-bridge elasticity. The latter effect also has the advantage of it leading to better reproduction of the relationship between [MgATP] and velocity [17] than the assumption of linear cross-bridge elasticity. This uncertainty about the cross-bridge’s elastic properties is the major reason why we did not change any rate constants when attempting to increase V_0_ in our simulated data. The model results in this regard would be in the upper part of the normal range (cf. [8]) without changing any rate functions, especially if assuming that the cross-bridge elasticity exhibits a non-linearity similar to that proposed in [43]

### 3.6. Relation to Other Similar Models

It is of relevance to consider the presently optimized model in the context of other recent related models. This is carried out in Figure 11, which indicates that modifying related models ([5,85]), as seen here for the model of Rahman et al. [6], would improve their predictive capacity as well. This would allow for optimized versions of these models instead of the present one to be used when more appropriate, e.g., for simpler computations and for situations with reduced complexity (Månsson, 2016) [5], or to take into account a range of effects of varied [Pi] values [85].

## 4. Materials and Methods

### 4.1. Modelling—General

Our bottom-up mechanokinetic models relate actomyosin interaction kinetics, actomyosin elastic properties, and other basic molecular properties to the steady-state contraction of ensembles of myosin motors in muscle and in vitro. The integration of actomyosin biochemistry with elastic and structural cross-bridge properties follows the formalism in [2], with details described more recently [4,5,9,88].

All model states and parameter values are defined in Figure 1 and in Table 1 and Table 2. The myosin (M) and actomyosin (AM) states have either substrate (ATP; T) or products (ADP, D; inorganic phosphate, P or Pi) in the active site. The strongly actin-bound states with ADP in the active site are of different biochemical and structural types (Figure 1). The state before the power stroke is indicated by the subscript “L” (low force), and a second AMD state after the power stroke is indicated by subscript “H” (high force). Finally, one AMD state (denoted AMD/AM) without a subscript in Figure 1 is assumed to follow the AMD_H_ state in the cycle after a second small sub-stroke that confers strain sensitivity to the ADP release. The AMD state is assumed to have an open nucleotide pocket from which ADP is very rapidly released, making the state in rapid equilibrium with the rigor (AM) state. On this basis, and because of the lack of major structural changes in connection with the ADP release, the AMD and AM states are lumped together into an AMD/AM state. Standard model parameter values (Table 1 and Table 2) for initial simulations are from independent solution biochemistry and single-molecule mechanics experiments [9] to as great of an extent as possible. The model structure and parameter values (Figure 1; Table 1 and Table 2) are similar to those in [6], with modifications as suggested in [7,8,9]. Most importantly, the rate constant for transition from the AMDP_PP_ to the AMD_PiR_ state was increased from 1000 s^−1^ in the original model [6] to 3000 s^−1^ in order to the avoid effects on V_0_ under some conditions. Simulations of FV data at low temperatures (22 °C or 5 °C) were performed using the parameter values in Table 3 (cf. [6] and references therein). Free energy differences *(*Δ*G_AMDP-AMDL_,* Δ*G_AMDL-AMDH_,* and Δ*G_AMDH-AM_*) between states are given in units of k_B_T (~4 pN nm), where k_B_ is the Boltzmann constant, and T is the absolute temperature. Moreover, simplifying assumptions of the modelling include a uniform distance (*x*) distribution between the myosin heads and the closest myosin binding site on actin (cf. [1,2,40]), with only one myosin head available for binding to a given actin site. Finally, we assume independence of the two heads of each myosin molecule and linear (Hookean) cross-bridge elasticity (*k_s_* = 2.8 pN/nm). 

The rate functions exhibit strain dependence, which is expressed in terms of the variable *x*. The latter is formally defined as the distance between a myosin head in the rigor (AM/AMD) state and the nearest binding site on actin so that the elastic strain in the rigor cross-bridge is zero at *x* = 0 nm. The transition in the cycle from the weakly bound AMDP state specifically to the first stereospecifically attached pre-power-stroke (AMDP_PP_) state is governed by the rate function: (1)$$k_{\mathrm{on}}(x)=k_{\mathrm{on}}\prime \;\exp(\Delta G_{\mathrm{on}}-k_{s}(x-x_{1}{)}^{2}/(4k_{B}T))$$
where ΔGon is the difference in free energy minima between the MDP and AMDP_PP_ states.

The reversal of the transition is governed by:*k_on-_*(*x*) = *k_on_’* exp(*k_s_* (*x* − *x*_1_)^2^/(4*k_B_T*))(2)

The transition into the subsequent Pi-release state (AMDP_PiR_) (19) and its reversal (20) are governed by: *k_Pr+_*(*x*) = *k_Pr+_’* exp(Δ*G_PiR_*/2 − (*k_s_*/2)(*x* − *x*_11_)^2^/(2*k_B_T*) + (*k_s_*/2)(*x* − *x*_1_)^2^/(2*k_B_T*))(3)
and *k_Pr−_*(*x*) = *k_Pr+_’* exp(Δ*G_PiR_*/2 + (*k_s_*/2)(*x* − *x*_11_)^2^/(2*k_B_T*) − (*k_s_*/2)(*x* − *x*_1_)^2^/(2*k_B_T*))(4)

Here, Δ*G_PiR_* is the difference between the free energy minima of the AMDP_PP_ and the AMDP_PiR_ states. 

Subsequently, Pi is assumed to be rapidly and reversibly released from the AMDP_PiR_ state in a strain-insensitive transition to form the AMD_L_ state. If the forward, first-order, rate constant is denoted *k_p+_*, the backward pseudo-first order rate constant (at constant [P_i_]) is given by: *k_p−_* = *k_p+_*[P_i_]/*K_C_*(5)
where *K_C_* is the dissociation constant for Pi-binding to myosin, and [P_i_] is the concentration of inorganic phosphate in solution.

The subsequent transition is the power stroke (a Huxley and Simmons [89] type of transition; see also [4]). The forward (7) and reverse (8) transitions are governed by: *k_LH+_*(*x*) = *k_LH−_*(*x*) exp(Δ*G_AMDL-AMDH_* + *k_s_*(*x* − *x*_1_)^2^/(2*k_B_T*) − *k_s_*(*x* − *x*_2_)^2^/(2*k_B_T*))(6)
and
*k_LH−_*(*x*) = 2000 s^−1^(7)
respectively. 

We next assume [6,8,88] that the transition from the AMD_H_ to the AMD state is governed by:(8)$$\;k_{5}\left(x\right)=k_{5}\left(x_{1}\right)\exp(∆G_{AMDH-AM}+\frac{k_{s}{\left(x-x_{2}\right)}^{2}}{2k_{B}T}-G_{AM}\left(x\right))$$
where
(9)$$G_{AM}\left(x\right)=\int_{x}^{x_{3}}F_{AM}\left(x’-x_{3}\right)dx{’}_{\;}/\;k_{B}T$$

In the case of linear cross-bridge elasticity, as assumed here, $F_{AM}\left(x’-x_{3}\right)$= *k_s_*(*x* − *x_3_*). Furthermore, as mentioned above, the states AMD and AM are lumped together into an AM/AMD state. 

The detachment rate function from the AM/AMD to the MT state can then be approximated by (cf. [6]):(10)$$k_{\mathrm{off}}x=\frac{k_{2}\left(x\right)k_{6}\left[MgATP\right]}{\frac{k_{6}}{K_{1}}+\left(k_{2}\left(x\right)+k_{6}\right)\left[MgATP\right]}=\frac{k_{2}\left(x\right)\left[MgATP\right]}{\frac{1}{K_{1}}+\frac{k_{2}\left(x\right)}{k_{6}}\left[MgATP\right]+\left[MgATP\right]}$$
with
(11)$$k_{2}\left(x\right)=k_{2}\left(0\right)exp\left(\frac{\left|F_{AM}\left(x-x_{3}\right)\right|\cdot x_{crit}}{k_{B}T}\right)$$

Here, *k*_2_(0) (*k*_2_ in Table 2) and k_6_ are rate constants for ATP-induced detachment from the AMT state at *x* = 0 and ADP dissociation from the AMD state, respectively. Because we assumed that [MgADP] ≈ 0, the latter transition (rate constant *k*_6_) is assumed to be irreversible. The parameter *K*_1_ is an equilibrium constant for MgATP binding to the AM/AMD state (Figure 1) and x_crit_ defines the strain sensitivity of *k*_2_(*x*) [5].

### 4.2. Solutions of Ordinary Differential Equations and Derivation of Simulated Muscle Properties from State Probabilities

Model state probabilities for muscle contraction at constant velocity, *v*, was modeled by solving the following differential equations (for all *j*,*k*): (12)$$\frac{da_{j}}{dx}=(\overset{n1}{\underset{k}{\sum }}k_{kj}\left(x\right)a_{k}\left(x\right)-\overset{n2}{\underset{k}{\sum }}\{k_{jk}\left(x\right)a_{j\left(x\right)}\}/v)$$
where *a_j_*(*x*) are the state probabilities for the MT (*j* = 6), MDP (*j* = 7), AMDP_PP_ (*j* = 1), AMDP_PiR_ (*j* = 2), AMD_L_ (*j* = 3), AMD_H_ (*j* = 4), and the AM/AMD (*j* = 5) states in Figure 1. The rate functions *k_kj_*(*x*) and *k_jk_*(*x*), given in detail above and represent transitions into n1 neighboring states and out of state *a_j_* (into *n*2 other states), respectively. The model simulations were performed by numeric solutions (Runge–Kutta–Fehlberg algorithm) of the differential equations using the program Simnon (cf. [7]). The observable variables were then calculated from appropriate state probabilities [40] by averaging over the inter-site distance (36 nm) along the actin filament. Using this approach, average force *<F>* (in pN) per myosin head (attached to actin or not) is given by: (13)<F≥ ∑15∫−9114ksaj(x)(x−xj)dx∑17∫−2214aj(x)dx
where the denominator represents summing over all states and *x* values, as seen in Figure 1. 

In order to ensure stability in the numerical computations, the value of any rate function (Equations (1)–(11)) was limited to a maximum (*r_max_*) of 100,000 s^−1^ for isometric contraction and to 1,000,000 s^−1^ for the fastest shortening velocities and to a minimum (*r_min_*) of 1 × 10^−6^ s^−1^. If any of the limits exceeded a certain value of *x*, the parameter value was set to either *r_max_* or *r_min_*. For other details regarding the implementation of the numerical integration method, please see [7]. 

### 4.3. Monte Carlo Simulations

Monte Carlo simulations were performed essentially as described in [5,6] using the Gillespie algorithm [90]. We originally [5,6] developed a method to simulate data from the in vitro motility assay. Therefore, the total number (*N*) of available myosin heads is defined by a surface motor density (ρ), an actin filament length (*L*), and the width (*d*) of a band surrounding the filament, where myosin motors may reach an attachment site on actin, i.e., *N* = ρ*dL*. As an approximation, we further assume (as in the differential equation implementation) that only one actin subunit per 36 nm of the actin filament is available for myosin head binding. 

Three versions of the simulations were run corresponding to 1. unloaded shortening, 2. pure isometric contraction, and 3. isometric contraction, with the imposition of iso-velocity shortening ramp upon attainment of the maximum isometric force. In the first of these running modes (unloaded shortening), the cross-bridge state distribution is allowed to move freely along the x-axis (defining the myosin head and actin site distance) to keep the total cross-bridge force equal to 0 pN. The resulting displacement per time then gives the unloaded shortening speed. In the second run mode (isometric contraction), the cross-bridge distribution evolves to with zero displacement over time between the actin site and the myosin head. Finally, in the third run mode, an approximate iso–velocity displacement (constant velocity displacement of the cross-bridge distribution along the x-axis) is imposed once the cross-bridges have reached their steady-state isometric distribution. 

In all running modes, the simulations started with the myosin heads being detached from actin and equilibrated between the MT and MDP state. Transitions were then stochastically selected, and inter-state transition times were stochastically determined using the Gillespie algorithm based on probabilities reflecting transition rates, as defined by Equations (1–11) (see [5,6,9] for details). This leads to a stochastic exploration of the state space defined in Figure 1. The population of different states at a given time is transformed to different force levels based on elastic and structural properties of the myosin cross-bridges, as defined by the free energy diagrams in Figure 1B. Average force values as well as average velocity values can be estimated from these simulated data, forming the basis for the construction of FV relationships.

In order to compare FV data for very large myosin ensembles between the approaches using the solution of a differential equation and Monte Carlo simulations, we assumed a low motor density (ρ) on the surface in our Monte Carlo simulation design together with very (unrealistically) long filaments. This was necessary to avoid competition between motors for each actin site. Systematic tests suggested that in order to obtain accurate and reliable large ensemble properties by the Monte Carlo approach, i.e., values similar to those obtained via the solution of differential equations, it was necessary to assume an ensemble size of >1000 myosin heads (*N*) (Appendix A). Whereas the simulated maximum isometric force per myosin head is virtually independent of *N*, V_0_ is reduced with reduced N, with saturation of *N* >~1000 at a V_0_ value very similar to that derived via the solution of differential equations. We further found that the appropriate condition to obtain *N* > 1000 was to have *L* > 400 µm and ρ < 250 µm^−2^. Combinations with ρ > 1000 µm^−2^ and *L* < 100 µm, i.e., >~1 head on average per available actin site, at 36 nm intervals along the filament failed to reproduce the data obtained via the solution of differential equations (inset Appendix A). We attribute this to competition between myosin heads to the actin sites.

In comparing the FV data from the Monte Carlo simulations to those obtained by solving differential equations, we used identical model structure (Figure 1) and parameter values (Table 1, Table 2 and Table 3) as well as very similar rate functions (Equations (1)–(11)) for the two methods. However, the Monte Carlo simulations assumed that the very fast power stroke transitions are represented by rapid equilibria, whereas the actual forward and backward transition rate constants were used to solve the differential equations. In the present implementation of the simulations, this had negligible effects on the results, as is clear from the very similar values of the FV data (see below) obtained by the Monte Carlo and differential equation approaches. Furthermore, in accordance with this view, several-fold increases in the power stroke transition rates had negligible effects on the simulation outcome.

### 4.4. Fit of Simulated and Experimental Data to Hill (1938) Hyperbolic Equation

In several of the figures below, both experimental and simulated FV data are fitted by the Hill (1938) hyperbolic equation [69] using non-linear regression (implemented in Graph Pad Prism, v. 9.3.1, Graph Pad Software, San Diego, CA, USA): (*F* + *a*) × (*V* + *b*) = *b* (*Fo* + *a*)(14)
where *a* and *b* are constants, *F* is force at a given velocity, *V* is the velocity, and *Fo* is the maximal isometric force when *V* = 0. Because *V*_0_ occurs for *F* = 0, *V*_0_ = (*b* × *Fo*)/*a*. 

## 5. Conclusions

Based on data for the molecular mechanisms and contractile function of the drug amrinone, we have arrived at an optimized version of the model from [6] (closely related to two other models [5,85]). This model is now in better agreement with recent estimates for myosin’s power stroke sub-components [44]. In the process, we elucidated parameter values of importance for determining the shape of the non-hyperbolic deviation of the force–velocity relationship at high loads. After increasing the attachment rate in the model by 2.5-fold to account for the maximum power output, the prediction of experimental force–velocity data was within the experimental uncertainty range (Figure 9) for both the huge ensembles of muscle and small myosin ensembles with >30 myosin heads. However, limitations of the model, e.g., the poor prediction of FV data for low N and the uncertainty of whether the myosin cross-bridge elasticity is linear [86] or non-linear [8,43] in muscle cells, need to be addressed before final use in assessing drugs and mutation effects.

## Figures and Tables

**Figure 1 ijms-23-12084-f001:**
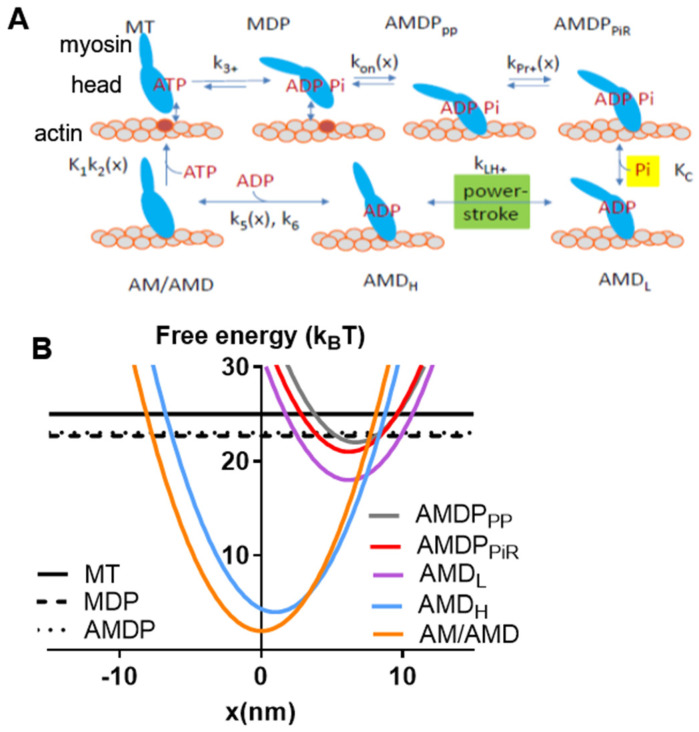
Key transitions and model states with characterization of major states. (**A**) Schematic illustration of myosin head states, including their coarse-grained structure in interaction with actin. The model states are encoded by the letters in the boxes, where A and M denote actin and myosin, respectively, and T, D, and P denote substrate and products, respectively (see further text). The subscripts PP and PiR denote a pre-power stroke state and a Pi-release state, respectively, as defined previously [6,46]. The subscripts L and H are defined in the text. Upper-case and lower-case letters for transition constants refer to equilibrium constants and rate constants, respectively. The argument x indicates the strain dependence of the constant. (**B**) Free energies of the states defined in A as a function of the strain variable, *x*.

**Figure 2 ijms-23-12084-f002:**
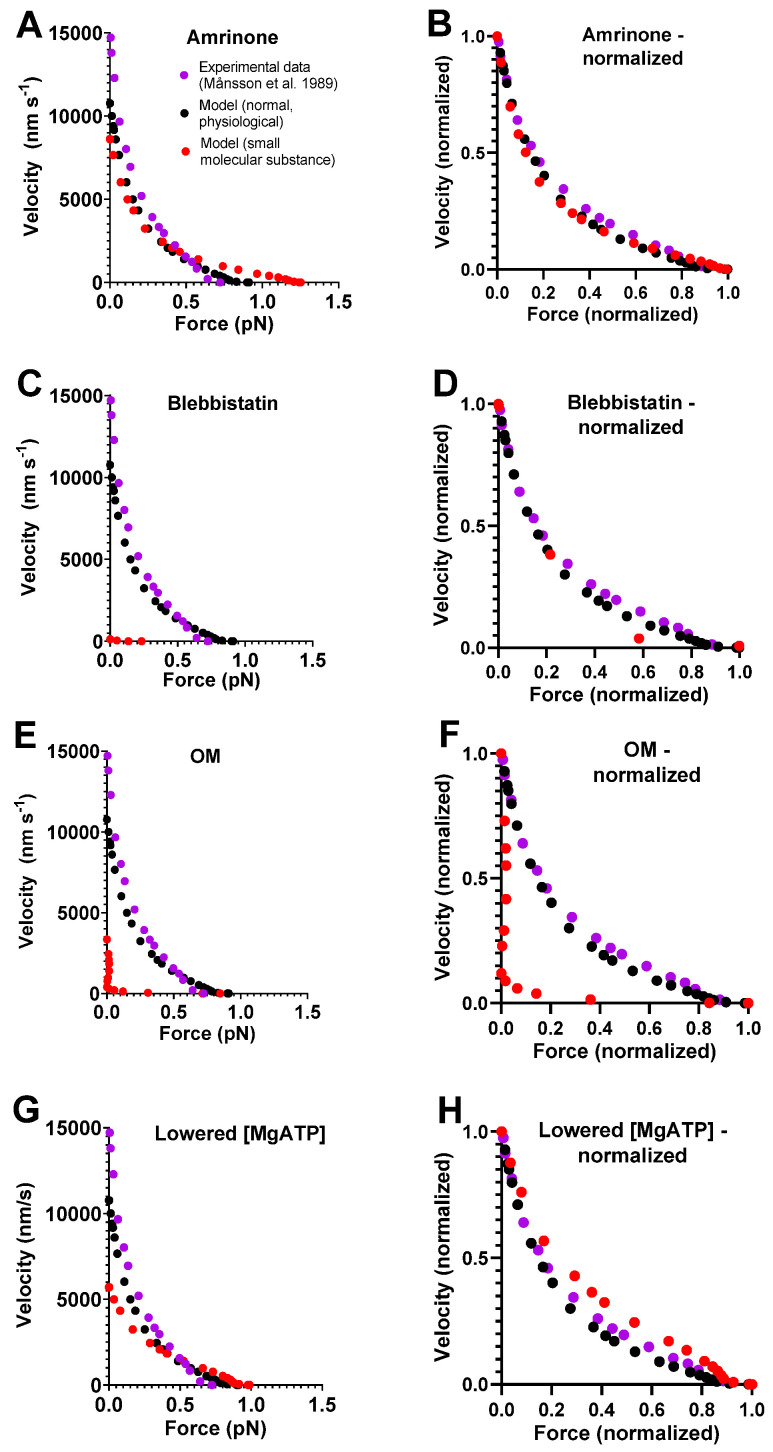
**Force–velocity data simulated using the model in Figure 1 with standard parameter values (Table 1 and Table 2).** (**A**) Simulated data from model under standard conditions (black) and modified conditions (as described in text) to account for the effects of 1–2 mM amrinone (red). The modeled data are compared to experimental FV data (purple) from [13] in the absence of any myosin-modifying compound. Maximum force given in pN per available cross-bridge (whether attached or not) for model data, whereas the maximum force for experimental data is normalized to exhibit the same maximum force as the model. (**B**) Data from A replotted after normalizing both force and velocity to maximum value under each condition. (**C**) Simulated data from A, but red symbols correspond to the effects of saturating the concentration of blebbistatin. (**D**) Data from C replotted after normalizing both force and velocity to maximum value under each condition. (**E**) Simulated data from A but red symbols correspond to effects of saturating concentration of OM. (**F**) Data from E replotted after normalizing both force and velocity to maximum value under each condition. (**G**) Simulated data from A, but red symbols correspond to effects of reducing [MgATP] from 5 mM to 100 µM. (**H**) Data from G replotted after normalizing both force and velocity to maximum value under respective conditions.

**Figure 3 ijms-23-12084-f003:**
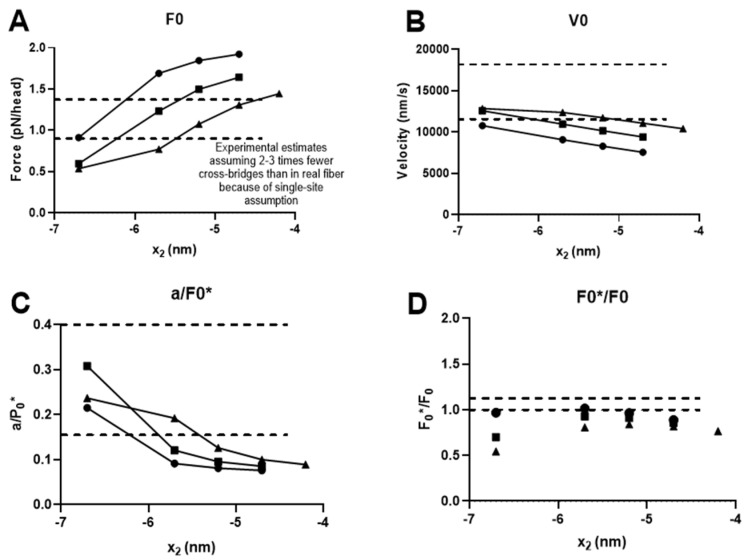
**Effects of changes in parameter value x_2_ at three different values of****Δ*G_AMDH-AM/AMD_* (at constant** Δ*G_AMDL-AMDH_ +*Δ*G_AMDH-AM/AMD_* = 16 k_B_T) on FV parameters and quantitative effects of 1–2 mM amrinone on *V*_0_ and *F*_0_. (**A**) Variations in simulated values (pN/(all cross-bridges; *N)*) of isometric force (*F*_0_) vs. *x*_2_ with Δ*G_AMDH-AMD_* of 2 k_B_T (black circles), 4 k_B_T (black squares), and 6 k_B_T (black triangles). The same symbols are used in the other panels. Normal range indicated by dashed lines corrected for the simplifying assumption in the model of one myosin binding site per actin target zone (at 36 nm interval) along the thin filament. The calculations of the normal range assumed 294 myosin heads per thick filament associated a cross-sectional area of 1.6 10^−15^ m^2^ and an isometric force per myofibrillar cross-sectional area of 450–500 kPa [58,59]. (**B**). Variations in simulated values of *V*_0 vs._
*x*_2_. Normal range indicated by dashed lines (cf. [8]). Same coding by symbols as in A. (**C**) Variations in simulated values of the ratio *a/F*_0_*** where vs. *x*_2_ where a is derived by a fit to Hill’s hyperbolic equation (see Materials and Methods) limited to force values below 80% of *F*_0_. The parameter *F*_0_*** is an estimate of F_0_ obtained by extrapolating the Hill equation to zero velocity. Normal range indicated by dashed lines. Same coding by symbols as in A. (**D**) Variations in simulated values of the ratio *F*_0_**/F*_0_ that is one estimate of the degree of deviation in the FV data from a hyperbola. Normal range indicated by dashed lines. Same coding by symbols as in A. Note that the ratio is highly sensitive to minor changes in the shape of the FV relationship. E. Simulated effects of amrinone (see text and below) on *F*_0_ and *V*_0_ with similar symbol coding as in A, but open symbols for *F*_0_ and filled symbols for *V*_0_. F. Simulated FV data after modifying x_2_ from −6.7 to −5.2 nm, Δ*G_AMDL-AMDH_* from 14 to 12 k_B_T, and Δ*G_AMDH-AM/AMD_* from 2 to 4 k_B_T but keeping all other model parameters at standard values (Table 1 and Table 2) for physiological conditions (black). Effect of amrinone (red) simulated by changing Δ*G_AMDH-AM/AMD_* to 1 k_B_T. Experimental data (purple) the same as in Figure 2.

**Figure 4 ijms-23-12084-f004:**
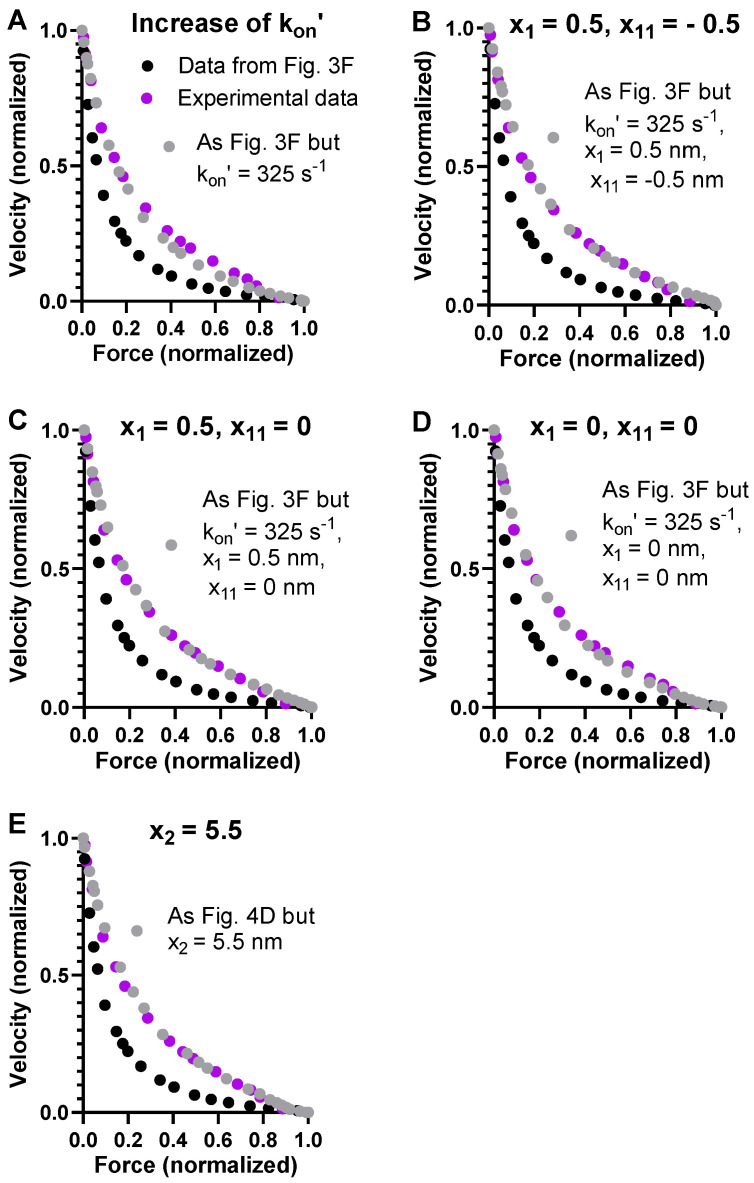
Modeled FV relationships with parameter values in Figure 3F showing effects of increased attachment rate constant and modified values of the parameters x_1_, x_11,_ and x_2_. (**A**) Effect of increased attachment rate constant *k_on_’* to 325 s^−1^ (grey symbols) compared to data simulated with standard parameter values (with *k_on_’* = 130 s^−1^; black-filled symbols) and experimental data used in other figures (purple). (**B**) Grey symbols refer to data modeled with attachment rate constant *k_on_’* = 325 s^−1^ but also with *x*_1_ (0.5 nm) and *x*_11_ (−0.5 nm), different from standard parameter values. Black and purple data are the same as in A. (**C**) Grey symbols refer to data modeled with attachment rate constant *k_on_’* = 325 s^−1^ but also with *x*_1_ (0.5 nm) and *x*_11_ (0 nm), different from standard parameter values and from their values in B. Black and purple data are the same as in A. (**D**) Grey symbols refer to data modeled with attachment rate constant *k_on_’* = 325 s^−1^ but also with *x*_1_ (0 nm) and *x*_11_ (0 nm), different from standard parameter values and from their values in B and C. Otherwise, color coding is the same as in A. (**E**) Grey symbols refer to data modeled with same parameter values as in D but with *x*_2_ = −5.5 nm. Black and purple symbols have same meaning as in (**A**) and in (**B**–**E**).

**Figure 5 ijms-23-12084-f005:**
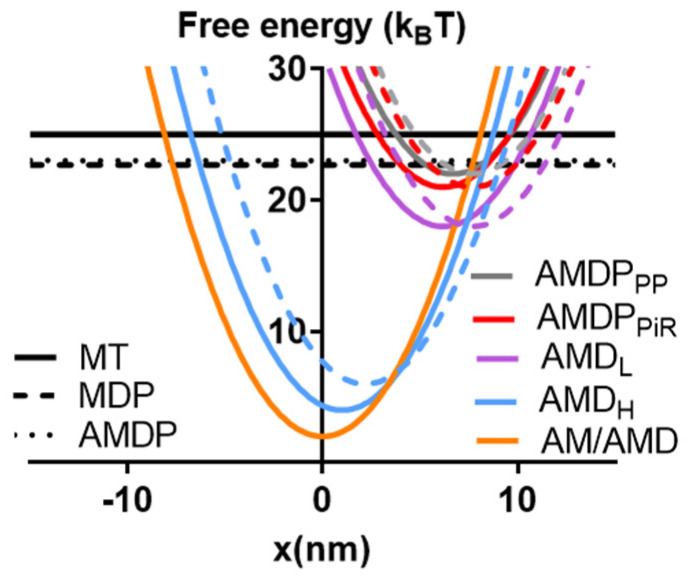
**Changes in minimum value and position of free energy diagrams in optimized model**. In the optimized model, we assumed different values than in Table 1 and Table 2 for the attachment rate constant (*k_on_’* = 325 s^−1^), but also (as indicated by the dashed curves) for the variables *x*_1_ (0 nm), *x*_11_ (0 nm), *x*_2_ (−5.5 nm), Δ*G_AMDL-AMDH_* = 12 k_B_T, and Δ*G_AMDH-AM/AMD_* = 4 k_B_T. The free energy diagrams with the standard parameter values are shown by full lines.

**Figure 6 ijms-23-12084-f006:**
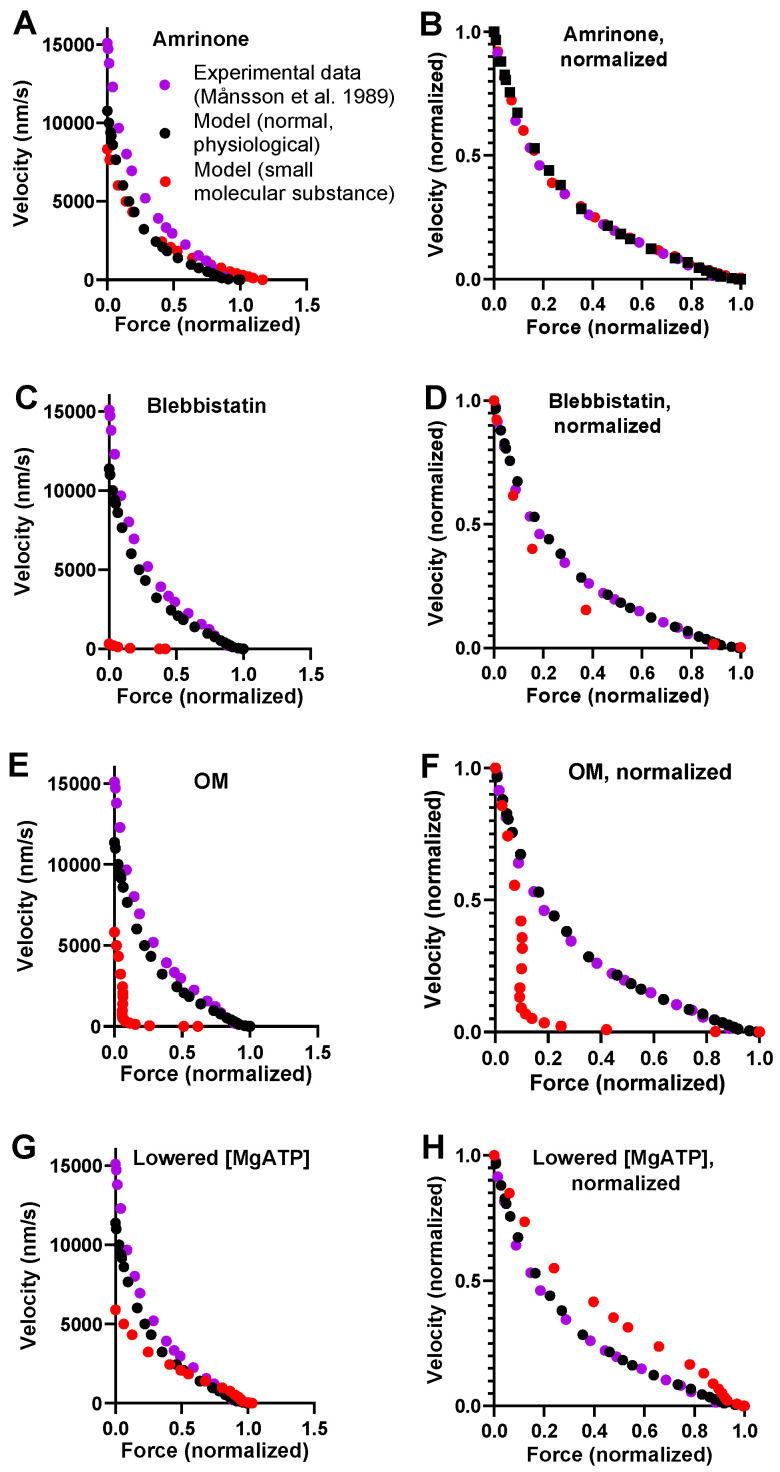
**Predictions of the FV relationship with effects of different compounds for the optimized version of the model (Figure 4E and Figure 5; Table 3).** (**A**) Simulated data from model under normal physiological conditions (black) and modified (as described in text) to account for the effects of 1–2 mM amrinone (red). The modeled data with amrinone, simulated by assuming a reduction in Δ*G_AMDH-AMD_* from 4 k_B_T to 1 k_B_T, are compared to experimental FV data (purple) from [13] in the absence of any myosin-modifying compound. Force given in pN per available cross-bridge (whether attached or not) for model data, whereas the maximum force for experimental data is normalized to exhibit the same maximum force as in the model. (**B**) Data from A replotted after normalizing both force and velocity to maximum value in each solution. (**C**) Simulated data from A, but red symbols correspond to the effects of the saturating concentration of blebbistatin simulated by assuming that this compound reduces *k_P+_’* from 3000 s^−1^ to 5 s^−1^. (**D**) Data from C replotted after normalizing both force and velocity to maximum value in each solution. (**E**) Simulated data from A, but red symbols correspond to the effects of saturating concentration of OM simulated by assuming that OM reduces Δ*G_AMDL-AMDH_* from 12 k_B_T to 0 k_B_T and increases Δ*G_PiR_* from 1 to 6 k_B_T. (**F**) Data from E replotted after normalizing both force and velocity to maximum value. (**G**) Simulated data from A, but red symbols correspond to effects of reducing [MgATP] from 5 mM to 100 µM. (**H**) Data from G replotted after normalizing both force and velocity to maximum value in each solution.

**Figure 7 ijms-23-12084-f007:**
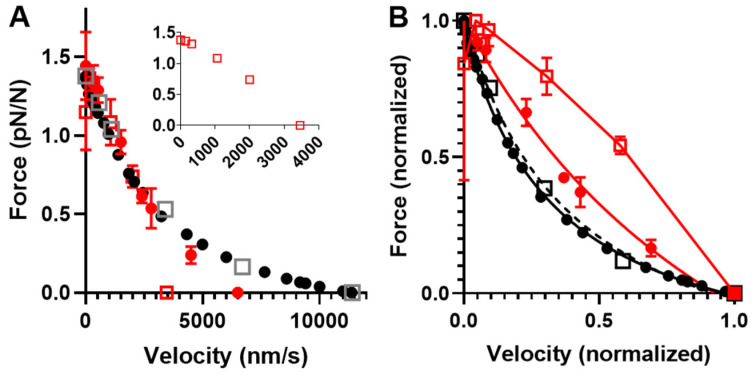
**Velocity–force data simulated using differential equations and Monte Carlo approach for ensembles of different sizes**. (**A**). Force vs. velocity based on solution of differential equations (Materials and Methods) in the state probabilities (black-filled circles; same as similar symbols in Figure 6) and Monte Carlo simulations assuming *N* = 3000 (open grey squares), *N* = 33–39 (filled red circles; mean ± 95% CI), and *N* = 16–19 (open red squares). Monte Carlo simulation data for *N* = 3000 based on 1 simulation run, whereas simulations for *N* = 16–19 and N=33–39 are based on 4 simulation runs per point except for isometric force (velocity= 0 nm/s), which is based on 28 simulation runs (*N* = 16–19) and 30 runs (*N* = 33–39). Inset: Data for *N* = 16–19 shown without error bars and assuming that the point for isometric contraction is identical to that at large *N* (consistent with overlapping 95% CI and data in Appendix A). (**B**). Data in A replotted after normalization of force and velocity to F_0_ and V_0_, respectively. Same color and symbol coding as in A. All data sets, except Monte Carlo simulation data for *N* = 16–19, fitted by Hill’s hyperbola (Equation (14)). Data points for *N* = 16–19 connected by lines for clarity. Note, virtually identical FV relationships (both in absolute and relative terms) for large ensembles whether data obtained by Monte Carlo simulations or solution of differential equations. Note further that the simulated FV relationship becomes less curved for low *N*, with loss of the hyperbolic shape for *N* = 16–19.

**Figure 8 ijms-23-12084-f008:**
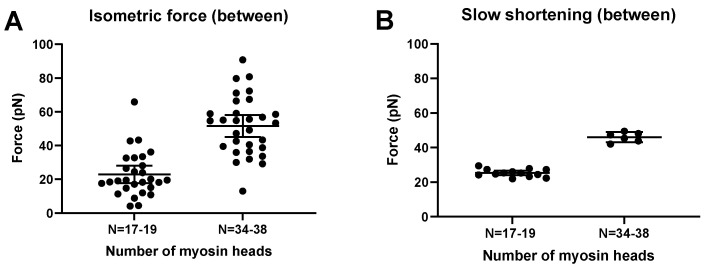
**Simulated data for isometric force and for force during slow shortening ramps.** (**A**) Isometric force from 28 simulation runs for *N* = 16–19 and 30 simulation runs for *N* = 33–39. Data given together with mean ± 95% CI. Each data point averaged over 15–55 data points during each simulation run. (**B**) Force during shortening ramps at 5% V_0_ for same conditions as in (**A**). Note that there is an appreciably smaller difference between simulation runs in B than in the case of the isometric force in (**A**).

**Figure 9 ijms-23-12084-f009:**
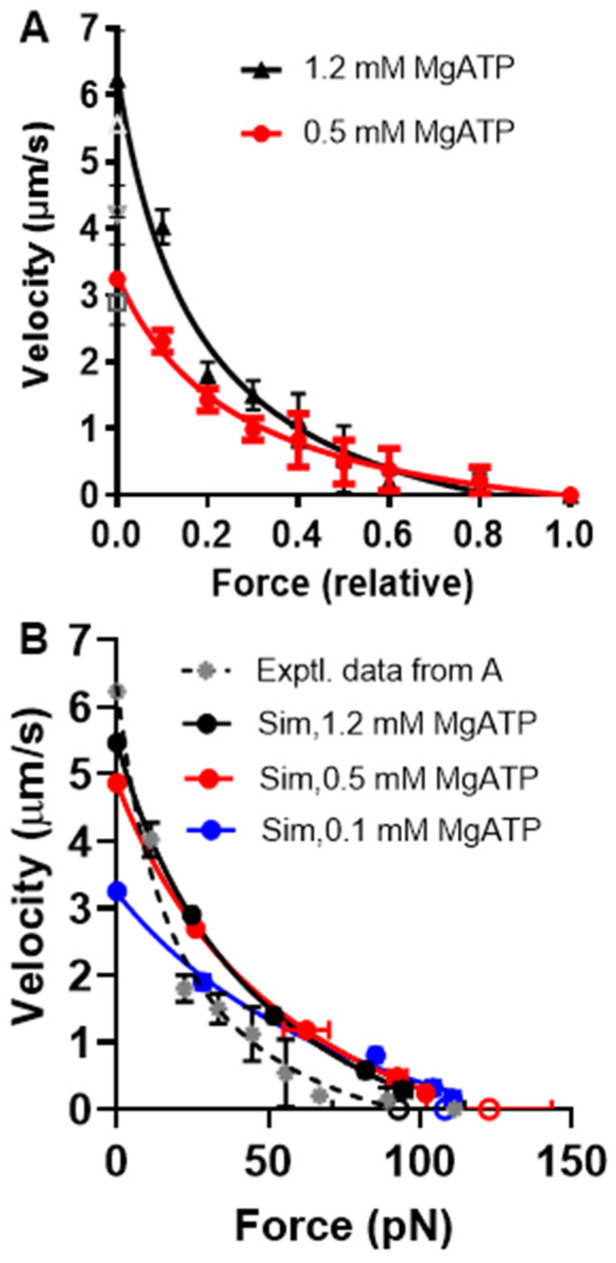
**FV relationship at different [MgATP] levels—experiments and simulations.** (**A**) Experimental data from Cheng et al. [70] from myosin filaments interacting with actin filaments at approximately 22 °C and either 1.2 mM (black) or 0.5 mM (red) MgATP. Force data normalized to F_0_. Measured F_0_ in experiments at 1.2 mM MgATP: 112.4 ± 11.0 pN/µm (*n* = 12) and at 0.5 mM MgATP: 54.5 ± 3.6 pN/µm (*n* = 8). Data shown as mean ± standard error of the mean (SEM). Experimental data fitted by Hill’s equation (see Materials and Methods) with *a/F*_0_*** = 0.27 at 1.2 mM MgATP and 0.40 at 0.5 mM MgATP. (**B**) FV data for 1.2 mM (black), 0.5 mM (red), and 0.1 mM (blue) MgATP simulated using Monte Carlo approach described in the text. Simulated data superimposed on experimental data (“Exptl.”, stars and dashed line) at 1.2 mM MgATP from A. Simulated data shown as mean ± SEM; *n* = 2 except for isometric force (open symbols), where *n* = 10, due to variability in average force between simulation runs. Simulated data (except point at *F*_0_) fitted by Hill’s equation with *a/F*_0_*** = 0.46 at 1.2 mM MgATP, 0.54 at 0.5 mM MgATP, and 0.74 at 0.1 mM MgATP.

**Figure 10 ijms-23-12084-f010:**
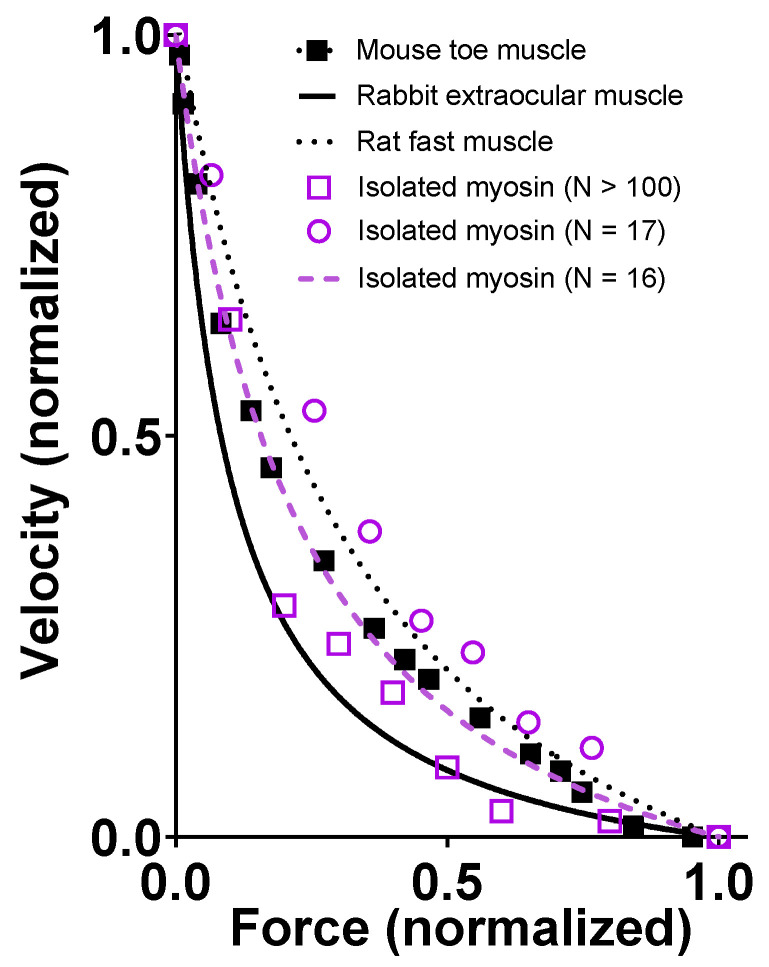
**Force–velocity data from different sources using living muscle cells and isolated actin myosin ensembles.** Data from living muscle from mouse toe [13] (black squares), rabbit extraocular muscle [49] (full line), and rat fast muscle [48] (dotted line). Other data are from isolated myosin ensembles from fast rabbit muscle interacting with one actin filament with either *N* = 17 [44] (open circles), *N* = 16 [66] (dashed line), or *N* > 100 [70] (open squares; same experimental data as in Figure 9). Data assembled by Månsson [9] either by re-plotting Hill equation fits or by measuring from the respective paper. Figure reproduced from [9], except for data from [70].

**Figure 11 ijms-23-12084-f011:**
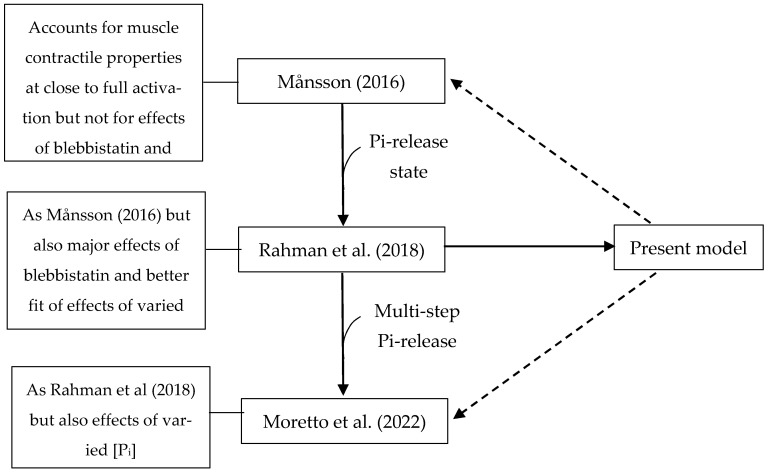
**History of model and implications for related models.** The model in Figure 1 is essentially the model of Rahman et al. (2018) [6] developed from the model of Månsson (2016) [5] to include a Pi-release state, allowing the model to account for most blebbistatin effects on actomyosin function and muscle contraction. The model of Rahman et al. [6] was then further modified with the inclusion of secondary Pi-binding sites outside the active site to account for a range of experimental observations at varied Pi levels, producing the model of Moretto et al. (2022) [85]. Importantly, the modifications of the model of Rahman et al. (2018) to produce the present optimized model should also be readily implemented (dashed arrows) in the models of Månsson (2016) and Moretto (2022).

**Table 1 ijms-23-12084-t001:** Parameter values ^a^ for model in Figure 1 determining the shape of free energy diagrams for simulation of contractile properties of fast mammalian muscle at 30 °C.

Parameter	Numerical Value of Parameter
*x* _1_ *(AMDP, AMDP_PP_)*	7.2 nm
*x* _1 1_ *(AMDP_PiR_, AMD_L_)*	6.7 nm
*x* _2_ *(AMD_H_)*	1.0 nm
*x* _3_	0 nm
Δ*G_w_ (MDP − AMDP)*	0 k_B_T
Δ*G_AMDP-AMDP-PP_* ≡ Δ*G_on_ (AMDP − AMDP_PP_)*	0.7 k_B_T
Δ*G _PiR_ (AMDP_PP_ − AMDP_PiR_)*	1 k_B_T
Δ*G_AMDPPiR-AMDL_ (AMDP_PiR_ − AMD_L_)*	k_B_T ln([P_i_]/K_C_)
Δ*G_AMDL-AMDH_ (AMD_L_ − AMD_H_)*	14 k_B_T
Δ*G_AMDH-AMD_**(AMD_H_ − AMD)*	2 k_B_T
Δ*G_ATP_*	13.1 + ln ([MgATP]/ ([MgADP][P_i_]) k_B_T
*k_s_*	2.8 pN/nm

^a^ The parameter values are primarily from two-headed myosin motor fragments from fast skeletal muscle from rabbit at 30 °C, ionic strength 130–200 mM, pH 7–8. For further details, see [6,7,8].

**Table 2 ijms-23-12084-t002:** Parameter values ^a^ for model in Figure 1 defining rate functions and kinetic constants for the simulation of contractile properties of fast mammalian muscle at 30 °C.

Parameter	Numerical Value of Parameter
*k_+_* _3_ *+ k_−_* _3_	220 s^−1^
*K* _3_	10
*k* _−5_	2000 s^−1^
*Kc*	10 mM
*k_on_’*	130 s^−1^
*k_Pr+_’*	3000 s^−1 b^
*k_P+_*	10000 s^−1^
*k_LH+_*	6000 s^−1^
*x_crit_*	0.6 nm
*k* _6_	5000 s^−1^
Physiological [Pi]	0.5 mM
[MgATP]	5 mM
*K* _1_	1.7 mM^−1^
*k* _2_	2000 s^−1^

^a^ The parameter values are primarily for two-headed myosin motor fragments from fast skeletal muscle from rabbit at 30 °C, ionic strength 130–200 mM, pH 7–8. For further details, see [6,7,8]. ^b^ Note difference from model in [6], where the same parameter value was set to 1000 s^−1^ under control conditions. Here, it was necessary to assume a higher value in order to achieve the experimentally observed V_0_ without changing other parameter values from their literature data.

**Table 3 ijms-23-12084-t003:** New parameter values for parameters modified to final optimized model and effects of temperature on these and other parameter values ^a^.

Parameter	30 °C	22 °C	15 °C	5 °C
Δ*G_AMDL-AMDH_*	12 k_B_T	9.2 k_B_T	7.2 k_B_T	5.1 k_B_T
Δ*G_AMDH-AMD_*	4 k_B_T	4 k_B_T	4 k_B_T	4 k_B_T
*k_on_’*	325 s^−1^	168 s^−1^	121 s^−1^	62.5 s^−1^
*x* _1_	0 nm	0 nm	0 nm	0 nm
*x* _11_	0 nm	0 nm	0 nm	0 nm
*x* _2_	−5.5 nm	−5.5 nm	−5.5 nm	−5.5 nm
*k_+_*_3_*+ k_−_*_3_(Recovery stroke+hydrolysis)	220 s^−1^	87.9 s^−1^	39.4 s^−1^	12.5 s^−1^
*K* _3_	10	7.5	5.8	4
*k* _2_	2000 s^−1^	1207 s^−1^	776 s^−1^	413 s^−1^
*k_P+_’*	3000 s^−1^	1925 s^−1^	1305 s^−1^	750 s^−1^

^a^ Parameter values not given here are assumed to be identical to those given in Table 1 and Table 2. Temperature effects estimated as described previously [6].

## Data Availability

Data are given in the paper or the Supplementary material. Numerical values of the data will provided (e.g., in Excel documents) upon reasonable request.

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
