# Peer review of "Insights into Muscle Contraction Derived from the Effects of Small-Molecular Actomyosin-Modulating Compounds"

_ijms, 2022, doi:10.3390/ijms232012084_

Round 1

Reviewer 1 Report

In this manuscript, by adjusting values of several key parameters in the model proposed before for myosin II, the theoretical data can become better agreement with the prior experimental data about the effect of the small drug compounds, in particular amrinone, on the force-velocity relationship of muscle. Additionally, with the newly adjustable parameters, using Monte-Carlo approach the force-velocity relationship from small myosin ensembles was also studied. Although the studies are interesting to the researchers in the field of muscle myosin motors, I do not think that compared to the prior studies these are so large increments that can be suitable for publication in the general journal like IJMS. I suggest that the manuscript may be suitable for a more specialized journal. In the future submission to other journal, I suggest that the authors should consider the following issues.

1)  The meaning of the purple dots in Fig. 2 has not been mentioned in the legend.

2)  The text should be checked carefully. For example, the word ‘there’ should be ‘the’ in line 243 ‘… have reached there steady-state …’. The word ‘to’ is missing in lines 259 and 260 ‘Systematic tests suggested that in order obtain accurate…’. The second word ‘in’ in line 909 ‘…in better in agreement with…’ should be deleted.

3)  Should term ks(x-x1)2/kBT in Eqs. (1) and (2) be ks/2(x-x1)2/(2kBT)?

4)  In lines 209 and 210, the state with j = 5 was not mentioned.

Author Response

The response is attached in a separate file

Reviewer 2 Report

This article describes the modelling of ensemble myosin-actin interactions. Such models are important to enable the characterisation of drugs/inhibitors and myosin mutants within tissues but based on single molecule measurements.

I have no concerns with the work and I believe it is a very useful model/concept for the community. Nevertheless, the presentation of several figures needs to be improved:

Fig3 is blurred

Fig 4 is blurred (partially)

Fig 5 appears to be low resolution

Fig 7 is low resolution

Fig9 is low resolution

Fig11 - Lowest box has been cropped.

Author Response

Response is submitted as a separate file

Reviewer 3 Report

In this manuscript, the authors demonstrate a model that have been published in (Biophys. J. 2018, 115, (2), 386) can be used to qualitatively predict the effects of drug molecules and varied [MgATP]. And they optimized the model to account for the effect of amrinone.  The simulations agree well with the experiments data from literature and the authors addressed that the changes of model parameters which determine the amplitudes of the power-stroke play key roles in the shape of the Force-Velocity curve. The main issue of the current manuscript is the quality of all the figures, which do not meet the requirements of scientific paper from my view. Details are listed below.

1) The label of y axis and unit are missed in Fig 1B.

2) Fig2 shows the Velocity-Force relationship at different conditions. The legends are all missed in each subfigure. It might be helpful to arrange the sub figures in a more reasonable way to minimize the empty space on the side and it might be helpful to add some texts to the subfigures to indicate the conditions.

3) The Fig3 is very blurred and all the texts in the figure is unreadable. The legend of all the subfigures are missed again and it is hard for reader to know the meaning of different lines and data points. It is wired that a dash line is outside of the figure box. The Fig3E is very busy and it is hard to distinguish different data points and lines

4) Fig4 has 5 very similar sub figures and  legends are missed.

5) Fig5 is blurred. Legends in Fig6 and Fig7 are missed. Fig9 is blurred and legends are missed. Fig10 has no legend.

6) Fig11 does not display correctly and it is might not necessary because this can be explained in the manuscript.

Another issue is that some of the equations in the manuscript are not formatted correctly and some latex syntax are in the equations. Such as, Eqs (8) (9) (13).

Page4: “Upper case and lower case letters for transition constants, refer to rate constants and equilibrium constants, respectively” should be Lower case and upper case letters for transition constants, refer to rate constants and equilibrium constants, respectively”

Some issues in the supplement:

1) Fig.S2 is very busy and it is hard to read.

2) The highlight texts in page5 of supplement need to be replaced with the right number.

Author Response

(The authors gave the same response as above.)

Round 2

Reviewer 3 Report

The link to the supplement downloads the manuscript of docx rather than the supplement.